# Investigating Dynamics of the Spinal Cord Injury Adjustment Model: Mediation Model Analysis

**DOI:** 10.3390/jcm11154557

**Published:** 2022-08-04

**Authors:** Ashley Craig, Yvonne Tran, Mohit Arora, Ilaria Pozzato, James W. Middleton

**Affiliations:** 1John Walsh Centre for Rehabilitation Research, The Kolling Institute, Royal North Shore Hospital, Northern Sydney Local Health District, St Leonards, NSW 2065, Australia; 2Faculty of Medicine and Health, The University of Sydney, Sydney, NSW 2006, Australia; 3Macquarie University Hearing, Faculty of Medicine, Health and Human Sciences, The Australian Hearing Hub, Macquarie University, Sydney, NSW 2109, Australia; 4Spinal Outreach Service, Royal Rehab, Ryde, NSW 2112, Australia

**Keywords:** spinal cord injury, adjustment, mediation, rehabilitation, neurological injury, depression, vitality, mental health

## Abstract

Spinal cord injury (SCI) is a severe neurological injury that results in damage to multiple bodily systems. SCI rehabilitation requires a significant focus on improving adjustment to the injury. This paper presents a detailed description of the Spinal Cord Injury Adjustment Model (SCIAM), which clarifies how individuals adjust to SCI and contends that adjustment to SCI is a multifactorial process involving non-linear dynamic adaptation over time. Evidence supporting SCIAM is also discussed. Mediation analyses were conducted to test the mediator dynamics proposed by the model. The analyses tested the relationship between two moderators (self-care and secondary health conditions), mediators (two self-efficacy items and appraisal of quality of life or QoL), and positive versus negative vitality/mental health as outcomes. Results showed that higher self-efficacy and perceived QoL was related to greater independence in self-care and reduced negative impacts of secondary health conditions. This study supported the mediation role of self-efficacy and other appraisals such as perceived QoL in enhancing self-care and buffering the negative impact of health challenges. In conclusion, it is important to employ a holistic model such as SCIAM to conceptualise and increase understanding of the process of adjustment following a severe neurological injury such as SCI.

## 1. Introduction

Spinal cord injury (SCI) is a severe neurological injury that results in damage to multiple bodily systems and functions. It is often associated with secondary conditions including chronic pain [1,2], spasticity [3], urinary tract infection [4], bowel complications [5], insulin resistance and diabetes [6], sexual dysfunction [7], pressure injuries [8], cardiovascular and respiratory problems [9,10], fatigue [11,12], cognitive impairment, and low vitality/fatigue and mental health problems [13,14,15,16]. In addition, it can lead to reduced social autonomy [17] and greater challenges for gaining and sustaining employment [18], and there exists a high risk of weight gain and sleep disturbance [19,20,21]. Given these multiple potential adverse conditions, improving self-management/self-care is a crucial goal to be achieved for successful rehabilitation outcomes [17,22]. The process of adjusting to a SCI, however, is a dynamic ongoing process that can highly influence a person’s self-care ability and recovery outcomes. Thus, it is critical that the complex dynamics of how people adjust to SCI over time are clarified. Arguably, if these dynamics are better understood, it should result in improved clinical management during the inpatient rehabilitation phase as well as in the development of more targeted self-care strategies that patients can use after discharge into the community.

### 1.1. SCI Rehabilitation

Advances in SCI rehabilitation and improved management of the abovementioned secondary health conditions along with more reliable long-term medical follow-up have resulted in increased life expectancy in people with SCI [23,24] although still well below life expectancy of the general population [24]. Clearly, SCI rehabilitation is important, and it has been defined by the World Health Organization (WHO) as a “goal-oriented process” intended to not only restore as much as possible, any remaining physical function but also achieve the highest level attainable of physical, psychosocial, and economic independence [25]. Due to the complex nature of the injury and need for comprehensive intervention, SCI rehabilitation requires an integration of multidisciplinary health interventions including physical, occupational, and psychosocial interventions in addition to medical and nursing management [26]. Initially, multidisciplinary intensive rehabilitation is required, driven by the need for individuals with SCI to adjust to the impact of the loss of motor, sensory, and autonomic function at or below the level of the lesion in the spinal cord as well as to be able to identify and know how to manage any associated secondary conditions. The person with SCI will also have to re-learn routine daily living skills, including bowel and bladder management, mobility, dressing, preparing meals and eating, bathing, managing finances, and shopping. It is crucial that the self-management skills acquired earlier in the rehabilitation process are translated into the patient’s community setting. Regrettably, successful community reintegration can be impeded by many factors, resulting in poor adjustment [17].

### 1.2. Factors That Impact Adjustment Following SCI

For an individual with SCI, adjustment can be defined as responding adaptively to injury, by modifying their behaviour, thoughts, and personal circumstances in relation to the multiple factors associated with the SCI and consequent impairment, with the goal being to achieve an acceptable quality of life through recommencement of desired social roles [17]. At the outset, it is important to say that the majority of adults with SCI, up to at least 60%, exhibit resilient behaviour and adjust well to SCI [27]. Nevertheless, adults with SCI generally do report lower quality of life (QoL) across all physical and mental domains, as assessed by the Medical Outcomes Study 36-Item Short-Form Health Survey (SF-36) [12,28], including physical functioning and psychosocial domains such as vitality/fatigue, emotional and social function, and mental health [12,28,29]. QoL is diminished by factors that cause distress, such as fatigue and chronic pain [12,28], pressure ulcers [8], and being overweight [30], as well as by negative perceptions, such as poor self-efficacy [28]. Furthermore, it is not unusual for people with a SCI to experience grief and sadness as a reaction to the associated impairment and loss [31]. Grief is an expected reaction to a catastrophic injury and can lead to positive adjustment if resolved adequately [17,31].

While the majority do adjust well to SCI, a large minority do not. Up to 40% will experience difficulties adjusting to the challenge of living with SCI and will remain at risk, for example, of developing clinically elevated levels of psychological distress and psychological disorders [15,32,33]. Comorbid mental health problems are also increased, with a diagnosis of depression as well as highly likely to be diagnosed with suicidal ideation, alcohol/drug abuse, or anxiety disorders [15]. SCI also results in sexual disorders that can be a major barrier to positive adjustment and mental health in men and women that, if not addressed, will likely result in lowered self-worth, limited intimate relationships, and reduced confidence to participate socially [7]. Other barriers to adjustment exist [2,9,11,13,17,18,21]. Some of the major factors that impact adjustment are discussed briefly below. For instance, lack of consistent self-management/self-care behaviour associated with challenges such as depressive mood may lead to increased occurrence of poor adjustment, resulting in self-neglect problems (e.g., lack of self-care), such as urinary tract infections and pressure injuries [4]. Additionally, recent research has identified that cognitive impairment is a barrier to adjustment to SCI [13,14]. For example, a dual diagnosis of SCI and traumatic brain injury (TBI) is associated with increased problems following discharge from hospital [34], poorer functional outcomes and longer acute rehabilitation length of stay than if no TBI is present [34]. Even the presence of a mild TBI (mTBI) can be a problem. mTBI is often associated with slower uptake of self-managed behaviour and independent living skills [13,14]. However, TBI is not the only known cause of cognitive impairment in people with SCI. Other factors that increase the occurrence of cognitive impairment include learning difficulties prior to the injury, pre-morbid head trauma, chronic fatigue, sleep disorder (such as obstructive sleep apnoea), chronic/neuropathic pain, elevated anxiety and depressive mood, polypharmacy including narcotics and neuroleptics, alcohol and substance abuse, older age, and inflammation associated with neural damage [13,14,16,35]. Research has also shown that the presence of diagnosed cognitive impairment in adults with SCI resulted in greater occurrence of depressive mood after discharge from inpatient rehabilitation [13]. Catastrophizing is a cognitive bias and involves imagining that the worst possible outcomes will occur related to one’s action or the problems being experienced [36]. Research has shown that individuals who catastrophize about their pain are at substantially higher risk of experiencing major depressive disorder [36].

It is estimated that between 30–60% of adults with SCI have a sleep disorder [37], such as obstructive sleep apnoea (OSA), and those with tetraplegia and/or complete lesions are most at risk [21,38]. The presence of OSA may be associated with problems falling sleep as well as interrupted sleep, pain, and spasticity [38]. Adults with SCI also suffer excessive daytime sleepiness (EDS), a predisposition to fall asleep during the day [37]. EDS will increase the occurrence of sleep disturbance that in turn will increase the occurrence of EDS [37]. Having a sleep disorder and EDS affects adjustment through reduced productivity and capacity for maintaining employment and social participation [39]. Adding to the complexity, sleep quality in turn may be disrupted by secondary conditions such as chronic pain, while sleep disturbance will increase pain levels [40]. Sleep disturbance is also associated with increased prevalence of depression and anxiety [15,21].

Chronic pain plays a major role in SCI adjustment. It has been estimated that up to 80% of adults with SCI experience chronic pain as a secondary condition, including musculoskeletal and neuropathic pain [2]. The presence of chronic pain increases the occurrence of problems such as fatigue, depressive mood and anxiety, lower resilience, functional and mobility limitations, and greater dependence on pain medications, such as opiates [15,41], and is strongly associated with negative cognitive biases such as pain catastrophizing [36].

Chronic fatigue is another serious barrier to adjustment following SCI [11,12,42,43]. Fatigue is a mental and physical state that involves feelings of excessive chronic tiredness, exhaustion, anxiety, and lowered mood [11] and should be differentiated from the daytime sleepiness and tiredness associated with daily physical and mental exertion [11]. Problems associated with fatigue for adults with SCI include increased errors when carrying out daily tasks, confusion, reduced motivation, circadian rhythm disruption, and elevated risk of developing anxiety and depressive mood [11,12,42].

Autonomic nervous system dysfunction can be a serious problem for adults with SCI, especially for those with lesions above the mid-thoracic level and/or complete lesions. It has been found to be associated with sleep disorder, EDS, and fatigue [37,44]. In cervical and thoracic lesions at T6 and above, disruption to sympathetic nervous system pathways occurs, resulting in an imbalance of sympathetic and parasympathetic activity. This can result in serious cardiovascular problems such as uncontrolled hypertension, referred to as autonomic dysreflexia (AD) [44], a life-threatening problem involving excessive reflex activity of the isolated sympathetic nervous system below the level of injury due to a nociceptive stimulus with vasoconstriction and unregulated rapid rise in blood pressure. AD can result in cardiac arrhythmias, seizures, intracranial haemorrhage, and even death. Autonomic nervous system dysfunction can also negatively influence respiratory control, thermoregulation, bowel, bladder, and sexual function [44]. Consequently, autonomic nervous system dysfunction is a substantial challenge for successful adjustment to SCI [37].

Finally, if a person with SCI experiences limitations and barriers to social participation and engagement, this can pose a major barrier to adjustment [17]. Research has shown that almost half of adults with SCI living in the community reported significant problems involving restricted social participation and social support, decreased mobility, and lowered access to employment and educational facilities [45]. The resulting effect will be an increased occurrence of mental health problems and self-neglect [17]. As discussed above, multiple factors contribute to poor adjustment following SCI, and arguably, these factors have the potential to combine, resulting in complex barriers to adjustment. A model that clarifies these processes involved in adjustment would result in an improved understanding of adjustment that may well lead to improved rehabilitation outcomes.

### 1.3. The SCI Adjustment Model (SCIAM)

The SCIAM was developed to clarify the process of adjustment to SCI [22]. It maintains that adjustment to SCI is a complex and dynamic process given potential interactions occur between physical impairment, associated secondary conditions, and psychosocial factors [22,46]. SCIAM endeavours to explain (and by necessity simplify), how such complex non-linear processes occur and how, in the course of time, ongoing life events and challenges influence ongoing adjustment. For example, it was shown that adults with SCI were more ready to change (reduce alcohol consumption) soon after the SCI, illustrating time complexity [47]. The passage of time is itself dynamic and non-linear and must be considered in any understanding of adjustment after SCI. As a further example of the complexity of time, individuals with SCI have been shown to become more depressive 12 months post injury when compared to their mood around 6–8 weeks post SCI when in the rehabilitation phase if they have cognitive impairment [13].

SCIAM [22] is grounded in the biopsychosocial model, integrating aspects of the health belief model, the response shift theory, and the transactional model of stress and coping [22]. The biopsychosocial model describes health status as an outcome of an interaction of biological, psychological, and social processes [48]. Features of the health belief model include, for example, that health is a product of peoples’ perceived barriers and susceptibilities and that multiple interactive factors contribute to health status [49]. Additionally, the response shift theory was also incorporated into SCIAM; that is, constructive change is believed to be driven by a “response shift”; that is, SCIAM incorporates a process of re-evaluation of values and beliefs about life choices and decisions [50]. Finally, SCIAM adopted aspects of the transactional model of stress and coping that suggests a person’s capacity to adjust is a transaction between themselves and their environment. In SCIAM, this transaction involves an appraisal and re-appraisal process of challenges experienced [22,51,52]. Figure 1 shows a simplified version of SCIAM.

The goal of SCIAM is to explain processes involved in adjustment with an aim of improving rehabilitation outcomes. As argued above, the time factor is important in SCIAM. It is expected that there would be a temporal influence from pre-injury, emergency, acute care, and inpatient rehabilitation through to social/community post-injury factors for adjustment status. For example, adjustment will be influenced by feelings of confusion, shock, disbelief, pain, anger, anxiety, grief, sadness, and helplessness experienced at various times in the post-injury timeline [22].

As shown in Figure 1, SCIAM explains adjustment in terms of moderators, mediators, and outcomes. A *moderator* is a factor that influences the strength or direction of a relationship between two other variables. A *mediator* explains the mechanism or process that influences a relationship between a moderator and an outcome variable by the addition of a third hypothetical variable, known as a mediator variable [53,54]. As an example, in SCI, time since injury acts as a moderator given it has been shown to strengthen the relationship between pain and mental health; that is, greater time since the SCI is associated with less pain and better mental health [53]. Self-efficacy acts as a mediator, as it has been shown to influence the relationship between a target moderator variable (e.g., pain) and an outcome variable (e.g., depressive mood). By way of explanation, pain and depressive mood generally have a positive association; that is, higher levels in one will result in higher levels of the other. However, self-efficacy has been shown to mediate this relationship so that higher levels of self-efficacy result in lowered pain and lowered depressive mood and vice versa [53]. It is important to note that in a mediation process, a mediator such as self-efficacy explains or at least partially explains how an internal psychological process influences moderator and outcome variables. This occurs through a mediating process that involves appraisals of a situation leading to coping strategies employed in response to the person’s appraisals [22].

In SCIAM, moderators are divided into three categories: (i) biological and medical factors such as age, sex, level of injury, completeness of the injury, pre-injury brain injury, secondary health conditions (e.g., bladder and bowel problems), and medications; (ii) psychological factors that include pre-injury mental health, personality, general self-efficacy, trait self-esteem, sexual relationships, caregiver factors, and cognitive capacity; (iii) social and environmental factors: social factors include stereotypes, policy frameworks, legal and political factors, compensation, social support, religious views, family relationships, and employment, while environmental factors include housing, accessibility, assistive technology, and health services availability. All these moderators can influence adjustment outcomes by strengthening or weakening outcomes, and they potentially interact with each other to alter outcomes.

In SCIAM, mediators involve appraisals (e.g., perceived susceptibility, perceived threat, and perceived ability to cope, often assessed using appraisal/self-efficacy measures) and coping strategies that can be adaptive (e.g., problem solving a challenge) or maladaptive (e.g., abuse of a substance, avoidance). Following the success or failure of a coping strategy after an appraisal, re-appraisal of the situation usually occurs, leading to no further action if all is well. If the problem remains unresolved, re-appraisal occurs, leading to additional adaptive coping or maladaptive coping strategies, such as negative emotional reactions and destructive behaviour. The relationship between mediators and moderators is believed to be dynamic, involving synergistic interactions [55], and as shown in Figure 1, SCIAM characterizes the process of mediation as incorporating both appraisal/re-appraisal and developing a coping strategy as the “engine room” of adjustment [22]. This is because it is believed this process contributes very substantially to the process of adjustment, empowering the person to adapt by employing appropriate appraisals and coping strategies. Self-efficacy is a recognized mediator [53]. It involves a person’s appraisals, expectations, and beliefs about executing a future task successfully [53] and has been shown to be a mediator of health outcomes [53,56,57]. The appraisal–reappraisal and coping strategy responses will also be influenced by moderators at various time periods, adding a further complexity that the model attempts to explain [58]. 

### 1.4. Evidence Supporting SCIAM

Evidence supporting the influence of moderators on adjustment is accumulating. Prospective research has clarified the role of moderators by following adults with SCI from admission to intensive rehabilitation through to discharge from intensive rehabilitation and up to 12 months post injury when living in the community [13,15,17]. Moderators that have been found to influence adjustment positively, in this case, perceived social participation, included younger age, fewer severe secondary medical conditions, and social support [17]. In the same sample of adults with SCI, moderators that were associated with poor adjustment, in this case depressive mood, included fewer years of education, the presence of pre-morbid psychiatric/psychological interventions, cognitive impairment, elevated anxiety at discharge from intensive rehabilitation, and greater severity of medical complications [15]. Research has rarely found injury characteristics such as the level of the lesion (e.g., paraplegia versus tetraplegia) or completeness of the lesion to influence adjustment significantly [15,17].

Recent research has studied how mood (a measure of adjustment) changes as a function of the moderating factors cognitive impairment and time since injury [13]. Mood was measured soon after admission to SCI rehabilitation, just before discharge from rehabilitation into the community, and 12 months post injury when participants were living in the community [13]. Cognitive impairment was assessed at the time of admission to rehabilitation. The adult patients with SCI were divided into those with impairment versus those with normal cognitive performance. Findings indicated that cognitive impairment interacting with time (moderators) had a significant influence on adjustment (i.e., mood). However, the moderating effect was complex, whereby depressive mood in those with cognitive impairment was no different from those with normal cognitive performance during inpatient rehabilitation but increased substantially in those adults who had cognitive impairment at 12 months post injury [13]. This was an important finding, as it highlighted the vulnerability of adults with SCI who have cognitive impairment when living in the community. It was concluded that the increase in depressive mood was associated with greater challenges in managing life due to their cognitive impairment after discharge into the community, and this is most likely because they had difficulty understanding important issues such as maintaining adequate social support or seeking medical input, resulting in their personal resources being stretched and tested and increasingly feeling overwhelmed and helpless. Adjusting to life post rehabilitation is difficult enough for those with normal cognitive function, and it will certainly be more distressing for someone who has difficulties with cognitively processing information [14]. In additional research conducted with the same participants, pain intensity (a moderator) was shown to influence pain catastrophizing (which is associated with adjustment), with higher pain intensity likely to result in higher pain catastrophizing [36]. The clinical implications of this research on adjustment suggests that SCI rehabilitation should be guided by the results of consistent and repeated psychological and cognitive testing and that additional attention should be targeted towards improving mood and pain in those who are vulnerable as well as improving skills of those with cognitive impairment before they transition into the community.

Directed regression analysis (or path analysis) was conducted with data from 70 adults with SCI who were living in the community [53]. The relationship of socio-demographic variables that act as possible moderators on chronic pain (moderator) and on mood (a proxy for adjustment) was calculated. Age, sex, years of education, level and completeness of lesion, presence of TBI, age at injury, and taking medications were not found to influence significantly chronic pain or negative mood states. However, time since injury did influence pain and mood. The directed regression analysis showed that less time since the injury was associated with a small but significant increase in chronic pain, which in turn resulted in increased depressive mood [53]. The converse was also confirmed. This indicates that moderators such as time since injury and chronic pain have a reciprocal influence on each other and a cumulative impact on adjustment over time.

Evidence for the influence of mediators on adjustment is also accumulating. As argued above, self-efficacy is considered to be an important mediator [22,27]. In the abovementioned regression analysis, self-efficacy, assessed by the Moorong Self-efficacy Scale (MSES) [56], was shown to mediate the relationship between pain and mood [53]. For example, the relationship between pain and depressive mood decreased significantly when self-efficacy was added to the regression equation. That is, self-efficacy buffered the negative effect of chronic pain on mood in the adults with SCI. Those with low self-efficacy were more likely to experience higher levels of chronic pain and clinically elevated depressive mood.

These results suggest that the following is happening: Those who believe their chronic pain is beyond their ability to manage (i.e., low self-efficacy) will more likely experience poor adjustment (e.g., elevated depressive mood). Alternatively, chronic pain may act to weaken self-efficacy, leading to negative appraisals (“I can’t do this anymore”), contributing further to depressive mood and poor adjustment. Robust self-efficacy is likely to act to reduce levels of chronic pain and depressive mood (“this is difficult, but I can manage it”). These findings emphasize the valuable role of enhancing self-efficacy in people with SCI during rehabilitation and that self-efficacy needs to be included in assessment as a sensitive prognostic rehabilitation outcome variable [55,56,57,58].

Using strength of social participation as an adjustment outcome, independent research investigated the validity of the mediating process in SCIAM [59]. It employed a cross-sectional design with individuals with SCI living in the community. They confirmed that higher self-efficacy was significantly associated with greater social participation [59]. A path analysis and significance of fit was then used to test the double-mediator process on adjustment (perceived participation). Results partially supported the proposed double-mediating mechanism in SCIAM. Other prospective research has found that self-efficacy influences adjustment (assessed by resilience) at discharge from inpatient rehabilitation and after 12 months post SCI [60]. We present further evidence in support of the SCIAM mediation dynamics by conducting mediation model analyses.

### 1.5. Aims and Hypotheses

The major aim of this paper was to test the dynamics of SCIAM. To achieve this, we ran a series of mediation analyses testing the relationship between the moderators “self-care” and “secondary health conditions” and outcomes “positive” and “negative” “vitality/mental health”. The analyses were also designed to test the impact of mediators “self-efficacy” and “perceived QoL” on the impact of the moderators on the outcome variables. It was hypothesized that robust mediators (e.g., higher self-efficacy) would enhance self-care behaviour, lower the negative impact of secondary health conditions (e.g., chronic pain), and as a consequence, improve vitality and mental health.

## 2. Methods

### 2.1. Participants, Study Setting, Procedure, and Ethics

Participants included those living with SCI in the community in Australia [61]. Inclusion criteria included: (i) aged 18 years or over, (ii) a traumatic or non-traumatic SCI, (iii) at least 12 months post SCI, and (iv) competent in English. All participants completed the International Spinal Cord Injury (InSCI) community survey [61,62] between January 2017 to May 2019. Details of the development and content of InSCI are available [61]. The data reported in this paper were taken from the Australian arm of the InSCI (Aus-InSCI) community survey [61,62]. The Aus-InSCI survey questionnaire consisted of 193 self-report questions, with 125 common to all 22 participating countries, with additional 68 questions in the Australian module related to areas such as pain, fatigue, and sleep [61,62].

Human research ethics approval was granted by the Northern Sydney Local Health District HREC (HREC/16/HAWKE/495) and the Australian Institute of Health and Welfare Ethics Committee (EO2017/1/341). The study complied with national laws, and regulatory approval conformed to the Declaration of Helsinki. All participants provided informed consent. The survey was mailed in 2018 to eligible Australian participants who were able to complete the survey as a paper copy or online using unique login details for each participant. A total of 9617 Australian records were supplied for data linkage, and after removal of duplicate records (*n* = 1649), those deceased (*n* = 1645), and those not eligible (*n* = 398), a total of 5925 surveys were mailed. Of these, 1579 surveys were completed (a response rate of 26.6%). Table 1 shows socio-demographic and injury factors for the participants. Participants and non-responders have a similar sociodemographic profile [61].

### 2.2. Measures

Socio-demographic and injury-related details were included from the Aus-InSCI community survey [61]. The outcome variable for the models tested included nine items taken from the Medical Outcomes SF-36 Mental health and Vitality subscales, shown to be reliable and valid measures [63]. The SF-36 is a widely used psychometric tool for assessing perspectives on health-related QoL [63]. Four were positive-worded items from these two domains (e.g., “Have you felt calm and peaceful?”), and five were negative-worded items (e.g., “Did you feel worn out?”). As SCIAM accepts that adjustment can be positive or negative, the models were analysed employing a positive-worded adjustment outcome (positive vitality/mental health: 4 positive items) and a negative-worded adjustment outcome (negative vitality/mental health: 5 negative items). The outcome variable was labelled “vitality/mental health” simply because they were a mixture of the mental health and vitality SF-36 domains.

For the mediator variables tested in the models, we employed two parallel self-efficacy items “serially” mediated by a third mediator: appraisal of QoL. The rationale for this was based on the assumption that self-efficacy appraisals linked with more specific events may in turn involve an overall appraisal of one’s life quality. The self-efficacy variables in the models tested consisted of two items the authors believe adequately represent the self-efficacy construct. The first item is from the General Self-efficacy Scale (GSES): “I am confident that I could deal efficiently with unexpected events” [64]. The GSES is a widely used and validated self-efficacy scale [64]. The second item is from the Moorong Self-Efficacy Scale (MSES): “How confident are you that you can maintain good health” [56]. The MSES is a reliable and valid questionnaire [56]. The QoL item “How would you rate your quality of life in the last 14 days (1: very poor to 5: very good)” was taken from the World Health Organization Quality of Life-BREF scale (WHOQoL-BREF; https://www.who.int/tools/whoqol (accessed on 4 July 2022)). The WHOQoL-BREF has been shown to have acceptable reliability and validity in SCI [65].

The modified version of the Spinal Cord Independence Measure (SCIM)—Self-Reported version (m-SCIM-SR) was used to provide the first moderator variable, that is, independence in activities of daily living [66]. This involved using six items from the Self-Care section of SCIM, covering activities such as eating, washing, grooming, and dressing. Each item ranges between “total assistance required” to “totally independent”. The m-SCIM-SR has been shown to be a valid and reliable instrument [67]. The Spinal Cord Injury Secondary Conditions Scale (SCI-SCS) was used to provide the second moderator variable. SCI-SCS assesses self-reported occurrence and severity of secondary conditions in persons with SCI, including areas such as pain, autonomic dysreflexia, postural hypotension, circulatory and respiratory problems, and pressure sores/ulcers [68]. The Aus-InSCI survey included 14 of 16 items of the SCI-SCS. These 14 items were summed, and the score was used as a moderator factor to indicate severity of secondary conditions. SCI-SCS is a reliable and valid instrument [68].

### 2.3. Analyses

Mediation model analyses were conducted to explain underlying relationships between two variables (e.g., a moderator or independent variable and an outcome variable or dependent variable) by the inclusion of a third variable, known as a mediator variable [69]. A significant mediation model analysis result suggests that the moderator variable is influencing the outcome variable, while the mediator variable influences both moderator and outcome variables by modifying underlying mechanisms [69]. It is important in a mediation analysis to identify potential confounding factors that may have an impact on the moderator and outcome variables that are not of interest.

Because the data used to determine mediation relationships were cross sectional, implicit mediation analyses were not possible [70]. Therefore, a measurement-of-mediation design was employed using a statistical approach to establish mediation [71]. To manage bias, bootstrapping, which does not assume the relationship between variables are normally distributed, was used to test the null hypothesis [71], and the choice of moderator, mediator, and outcome variables was chosen based on plausible relationships within the biopsychosocial framework of SCIAM.

Partial least squares path modelling was conducted to test for direct and indirect relationships between moderators SCIM—Self-Care activities and secondary conditions with the outcome variables. Analyses were performed using the PROCESS procedure V3.4 in SPSS version 27 [71]. From PROCESS, model 80 was used for the mediation model. Non-parametric bootstrapping analysis was used to test the models in this study. Mediation was found to be significant if the 95% bias-corrected confidence intervals for the indirect effects did not include zero [72]. To assess how much of an effect was mediated through the indirect pathway, we calculated the mediation proportion, defined as the proportion from the indirect effect (the mediator) on the total effects, that is, the indirect effect divided by the total effect. All models were adjusted for the possible confounders of age, sex, injury level, and time since injury. To assess how much of an effect was meditated through the serial pathway, we calculated the mediation proportion, defined as the proportion from the indirect effect (the mediator) on the total effects, that is the indirect effect divided by the total effect [73].

Figure 2 shows the hypothesized mediation path model for SCIAM that was tested. The model includes nine direct paths (*a–i*) with the two (parallel) mediators (M1 and M2) that are serially antecedent to the QoL mediator (M3). “X” represents the moderator variable. “Y” represents the adjustment outcome variable. The model analyses included testing two outcomes: “positive vitality/mental health” determined from the 4 positive-framed SF-36 vitality and mental health items and “negative vitality/mental health” determined from the 5 negative SF-36 vitality and mental health items. The strength of the pathways is indicated by standardized regression coefficients (β), which are indicated on the pathways.

## 3. Results

### 3.1. Model 1: Self-Care and Vitality/Mental Health

Results for the model were based on 5000 bootstrapped samples and showed a significant total effect for the relationship between self-care and positive and negative vitality/mental health outcomes when mediated by self-efficacy and perceived QoL factors (b(Total) = 0.13, SE = 0.004, *p* = 0.002 and b(Total) = 0.11, SE = 0.004, *p* = 0.006, respectively). All five indirect pathways were found to be significant for both outcomes (see Table 2A,B for details). The direct effect between SCIM—Self-Care and positive and negative vitality/mental health was not significant (b(Direct)) = −0.003, SE = 0.003, *p* = 0.21 and b(Direct) = −0.02, SE = 0.003, *p* = 0.36, respectively), suggesting a full mediation for both outcomes and indicating that the relationship can be explained through serial pathways (see Figure 3 and Figure 4).

### 3.2. Model 2: Secondary Conditions and Vitality/Mental Health

Results for the model were based on 5000 bootstrapped samples. A significant total effect for the relationship between health conditions and positive and negative vitality/mental health outcomes when mediated by self-efficacy and perceived QoL factors was found (b(Total) = −0.04, SE = 0.002, *p* < 0.001 and b(Total) = −0.04, SE = 0.002, *p* < 0.001, respectively). All five indirect pathways were found to be significant (see Table 3A,B for details). The direct effect between secondary conditions and positive and negative vitality/mental health was significant (b(Direct)) = −0.01, SE = 0.002, *p* < 0.001 and b(Direct)) = -0.02, SE = 0.002, *p* < 0.001, respectively), suggesting a partial mediation for both outcomes and indicating that the relationship can be explained through serial pathways (see Figure 5 and Figure 6).

## 4. Discussion

The detail and evidence presented in the introduction supports SCIAM in that the process involved for a person adjusting to SCI is dynamic and non-linear and is contributed to over time by multiple factors [13,17]. Numerous factors have been shown to influence adjustment and include social participation and support; secondary complications such as sleep, pain, and weight; and sexual function, cognitive capacity, a younger age, and mental health, to name a few [7,15,17,19,20,21]. This extensive list demonstrates the difficulty of adjusting to SCI. Our focused review of adjustment also suggests that more controlled research is necessary, for example, prospective research that assesses relationships between factors from the acute stage after the injury to at least 1–2 years post injury, so that the complex dynamics involved in the process of adjustment can be thoroughly explored in a controlled manner.

SCIAM also proposes that different categories of factors will influence the adjustment process, such as moderators and mediators, both of which contribute directly or indirectly to positive or negative adjustment outcomes. It was argued that the mediator process is crucial for an individual to adjust positively. While we and independent researchers have provided some evidence supporting the importance of the mediator process for adjustment [53,59], again, further research is required to clarify protective and risk factors involved in how people adjust over time.

In response to this need, mediation analyses were conducted in an attempt to clarify aspects of the mediation process involved in adjusting to SCI. The first mediation model involved the influence of self-care activities on adjustment (positive and negative vitality/mental health), investigating the influence of parallel and serial mediators. It was concluded that self-efficacy and the appraisal of QoL fully mediated the influence of self-care activities on positive and negative vitality/mental health. This is understandable. If a person with SCI believes they have control of aspects of their self-care behaviour (e.g., “It takes time, but I can dress myself, even though I need some support”), then it is probable that this will enhance the likelihood of engaging in these behaviours, resulting in stronger vitality/mental health. Conversely, if an individual negatively appraises self-care activities as outside of their control (“Why even try, I cannot dress myself, I will always totally depend on others”), then it is likely that this will reduce engagement in self-care behaviour, potentially resulting in increased negative vitality/mental health. Self-care is well within one’s sphere of influence.

The second model involved the influence of secondary health conditions on adjustment. It was concluded that self-efficacy and the appraisal of QoL only partially mediated the influence of secondary health conditions on positive and negative vitality/mental health. This is also understandable. Developing a secondary condition such as bowel or bladder dysfunction, chronic pain, spasticity, or a pressure injury is dependent on factors that can be, to some degree, outside of an individual’s control. Therefore, the mediator process was only found to partially influence the relationship between secondary health conditions and vitality/mental health., This suggests that if an individual believes they have some control over the development of these health conditions, (e.g., “I need to take steps to reduce risk of bowel and bladder problems”), then it is probable that this appraisal will buffer and protect them from some of the negative effect of these conditions, resulting in stronger vitality/mental health [58,74]. Conversely, if an individual negatively appraises their risk of these health conditions as being outside of their control (“What is the point of trying to prevent problems with my bowels”), then it is probable that this will reduce healthy self-management, resulting in increased negative vitality/mental health. Partial mediation suggests it is less likely that a mediator such as self-efficacy will substantially buffer the negative effects of secondary conditions on adjustment.

There are certain limitations to consider. The data used to conduct the analyses were cross-sectional, and therefore, the findings do need to be replicated using implicit mediation analyses based on experimental design, allowing the manipulation of features of the relationships that implicate some mediators and not others [70]. However, participant numbers were large, and bias was managed using a bootstrapping technique, and the analyses were adjusted for possible confounders such as sex, age, and extent of the SCI. Further, given that the database contained no information on coping strategies that participants could employ to manage problems, we were unable to test the dual-mediation aspect of SCIAM. Lastly, the choice of an outcome variable was influenced by SCIAM presenting adjustment outcomes as either positive or negative. Therefore, we believed an acceptable approach to account for this was by investigating separate relationships between the moderators and mediators for two outcomes: positive and negative vitality/mental health.

## 5. Conclusions

We provide further supporting evidence for the potential benefits of the mediation process as described in SCIAM. We contend that the adjustment to SCI over time will be strengthened if strategies are employed that reinforce positive appraisals concerning self-management capacity. It is crucial that whatever self-management strategy is employed, it accentuates the importance of personal responsibility and realistic/optimistic thinking [17,33,36,53]. Strategies can range from encouraging specific self-management behaviours in bowel and bladder management, pain management techniques, enhancing assistive technology use, and employing cognitive strategies to encourage positive appraisals [36,46,47,57,58,59,60,75]. Finally, it is hoped this support for the crucial role and beneficial influence of mediators on adjustment following SCI will result in the introduction of structured rehabilitation processes, such as the development of psychosocial guidelines, to encourage consistent assessment of measures of mediators such as self-efficacy throughout the rehabilitation inpatient phase in addition to structured intervention strategies that enhance perceptions of self-responsibility and management. 

## Figures and Tables

**Figure 1 jcm-11-04557-f001:**
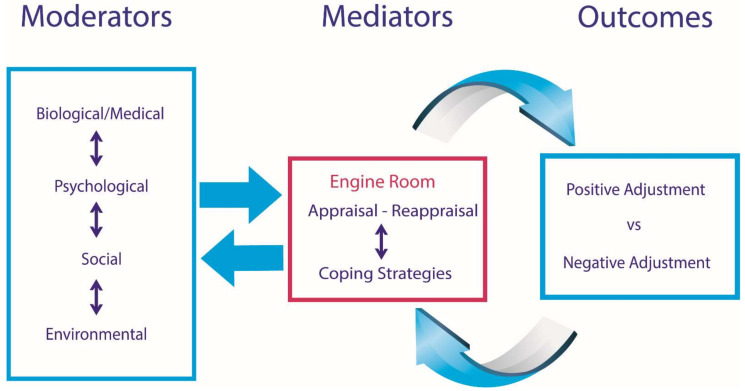
A simplified version of SCIAM (so that mediation analyses can be applied). Moderators will consist of multiple predisposing and modifying factors including age, sex, personality, injury severity, secondary conditions, religious and cultural views, employment and finances, policy and compensation, general self-efficacy and self-esteem, medications, and relationships. Moderators will influence mediators and vice versa. Mediators consist of appraisals and re-appraisals of life challenges and resultant coping strategies. There is a reciprocal influence between perceptions and coping strategies.

**Figure 2 jcm-11-04557-f002:**
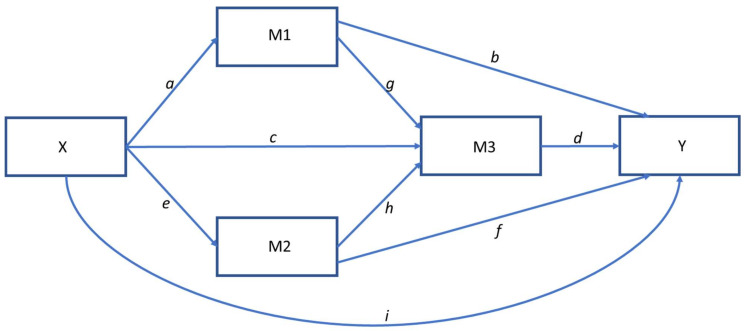
Hypothesized mediation path model for SCIAM. Model includes nine direct paths (*a*–*i*) with two parallel mediators (M1 and M2) that are serially antecedent to a third mediator (M3).

**Figure 3 jcm-11-04557-f003:**
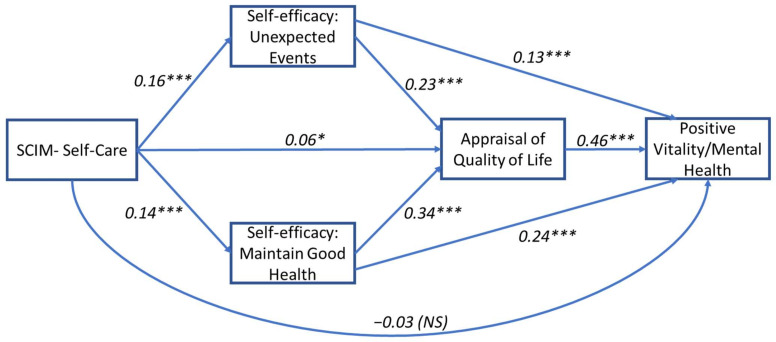
Mediation path model for the relationship between SCIM—Self-Care activities and positive vitality/mental health. Standardized regression coefficients (β) are presented for the direct paths. Model includes two parallel self-efficacy mediators (unexpected events and maintain good health) that are serially antecedent to a third mediator (self-appraisal of quality of life). Model is adjusted for age, sex, time since injury, and injury level. *** *p* < 0.001, * *p* < 0.05; NS, not significant.

**Figure 4 jcm-11-04557-f004:**
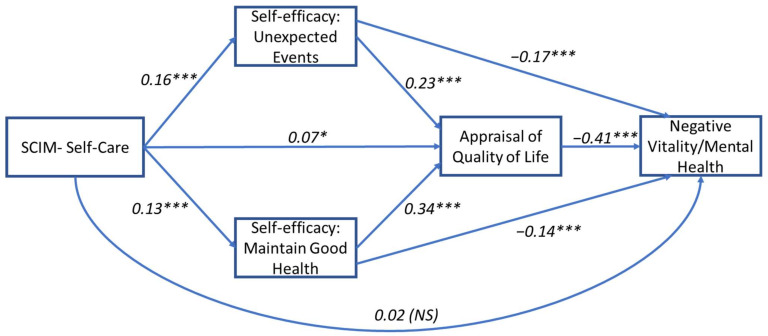
Mediation path model for the relationship between self-care and negative vitality/mental health. Standardized regression coefficients (β) are presented for the direct paths. Model includes two parallel self-efficacy mediators (unexpected events and maintain good health) that are serially antecedent to a third mediator (self-appraisal of quality of life). Model is adjusted for age, sex, time since injury, and injury level. *** *p* < 0.001, * *p* < 0.05; NS, not significant.

**Figure 5 jcm-11-04557-f005:**
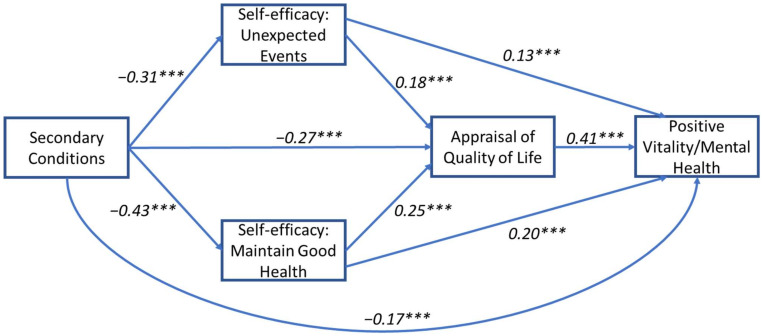
Mediation path model for the relationship between secondary conditions and positive vitality/mental health. Standardized regression coefficients (β) are presented for the direct paths. Model includes two parallel self-efficacy mediators (unexpected events and maintain good health) that are serially antecedent to a third mediator (self-appraisal of quality of life). Model is adjusted for age, sex, time since injury, and injury level. *** *p* < 0.001.

**Figure 6 jcm-11-04557-f006:**
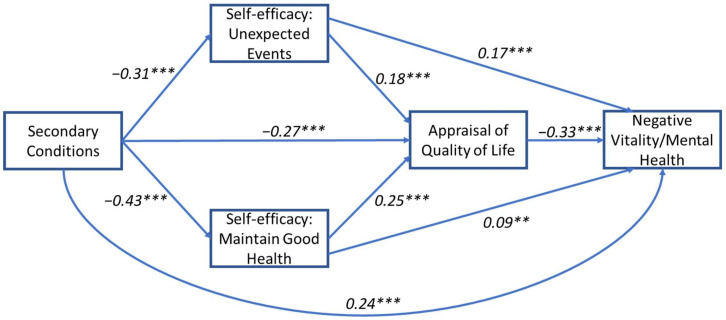
Mediation path model for the relationship between secondary conditions and negative vitality/mental health. Standardised regression coefficients (β) are presented for the direct paths. Model includes two parallel self-efficacy mediators (unexpected events and maintain good health) that are serially antecedent to a third mediator (self-appraisal of quality of life). Model is adjusted for age, sex, time since injury, and injury level. *** *p* < 0.001, ** *p* < 0.01.

**Table 1 jcm-11-04557-t001:** The demographics for the sample of 1579 participants.

Characteristics		Values
Age (years), mean (SD)		57.5 (14.4)
Years since injury (years), mean (SD)		17.2 (13.9)
Level of injury *	Paraplegia, *n* (%)	912 (57.8)
	Tetraplegia, *n* (%)	580 (36.7)
Completeness of injury **	Complete, *n* (%)	498 (31.5)
	Incomplete, *n* (%)	1042 (66.0)
Sex	Male, *n* (%)	1157 (73.3)
	Female, *n* (%)	422 (26.7)

* *n* = 87 (5.5%) missing ** *n* = 39 (2.5%) missing.

**Table 2 jcm-11-04557-t002:** Indirect effects for the indirect pathways explaining the relationship between self-care and positive vitality/mental health (**A**) and negative vitality/mental health (**B**). Bias-corrected bootstrapping was used. SC, SCIM—Self-Care; UE, self-efficacy for unexpected events; MGH, self-efficacy for maintaining good health; QoL, self-appraisal for quality of life; PVMH, positive vitality/mental health; NVMH, negative vitality/mental health.

Mediation Pathway	Standardised Indirect Effect	S.E.	95% CI	Mediation Proportion *
A				
SC > UE > PVMH	0.022	0.006	0.011, 0.035	0.23
SC > MGH > PVMH	0.033	0.008	0.018, 0.051	0.36
SC > QoL > PVMH	0.029	0.013	0.004, 0.055	0.31
SC > UE > QoL > PVMH	0.017	0.004	0.010, 0.025	0.18
SC > MGH > QoL > PVMH	0.022	0.005	0.012, 0.033	0.23
B				
SC > UE > NVMH	−0.027	0.007	−0.04, −0.01	0.33
SC > MGH > NVMH	−0.018	0.006	−0.03, −0.008	0.22
SC > QoL > NVMH	−0.027	0.011	−0.05, −0.005	0.33
SC > UE > QoL > NVMH	−0.014	0.004	−0.02, −0.008	0.18
SC > MGH > QoL > NVMH	−0.019	0.005	−0.03, −0.010	0.22

* Mediation proportions measure the proportion of the effect that is mediated. Note that this measure can be greater than 1 when there is an inconsistent mediation, that is, when the direct effect is an opposite sign to the indirect effect.

**Table 3 jcm-11-04557-t003:** Indirect effects for the indirect pathways explaining the relationship between secondary conditions and positive vitality/mental health (**A**) and negative vitality/mental health (**B**). Bias-corrected bootstrapping was used. SC, secondary conditions; UE, self-efficacy for unexpected events; MGH, self-efficacy for maintaining good health; QoL, self-appraisal for quality of life; PVMH, positive vitality/mental health; NVMH, negative vitality/mental health.

Mediation Pathway	Standardised Indirect Effect	S.E.	95% CI	Mediation Proportion *
A				
SC > UE > PVMH	−0.041	0.009	−0.06, −0.02	0.09
SC > MGH > PVMH	−0.088	0.013	−0.11, −0.06	0.19
SC > QoL > PVMH	−0.108	0.014	−0.14, 0.08	0.24
SC > UE > QoL > PVMH	−0.023	0.004	−0.03, −0.02	0.05
SC > MGH > QoL > PVMH	−0.044	0.006	−0.06, −0.03	0.10
B				
SC > UE > NVMH	0.054	0.010	0.036, 0.074	0.11
SC > MGH > NVMH	0.037	0.012	0.013, 0.062	0.08
SC > QoL > NVMH	0.090	0.013	0.066, 0.116	0.19
SC > UE > QoL > NVMH	0.019	0.004	0.012, 0.026	0.04
SC > MGH > QoL > NVMH	0.036	0.006	0.025, 0.048	0.08

* Mediation proportions measure that proportion if the effect that is mediated. Note that this measure can be greater than 1 when there is an inconsistent mediation, that is, when the direct effect is an opposite sign to the indirect effect.

## Data Availability

De-identified data are available upon request and with permission gained from the Aus-InSCI Community Survey National Scientific Committee.

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
