# Peer review of "Investigating Dynamics of the Spinal Cord Injury Adjustment Model: Mediation Model Analysis"

_jcm, 2022, doi:10.3390/jcm11154557_

Round 1

Reviewer 1 Report

The manuscript titled “Investigating the Complex Dynamics of Adjustment for Individuals with Spinal Cord Injury: A Narrative Review and Mediation Model Analysis” presents the application of the Spinal Cord Injury Adjustment Model that the authors developed previously to increase understanding of the complex process of adjustment following SCI. Although the authors performed a relatively comprehensive background review of the topics discussed in this paper, the reporting of the results and discussions suffers from insufficiency and redundancy. This reviewer suggests resubmission after major revision. Detail comments are below.

Sections 2 and 3: depending on the journals suggested structure, it is better to present these sections as subsections for the introduction i.e. 1.1, 1.2, etc.

3. Factors that impact on adjustment following SCI: in addition to factors described, sexual dysfunction should also be included since it poses a great impact on social interactions, self-worth and overall quality of life after SCI. Please include relevant references.

4. The SCI Adjustment Model (SCIAM): Depending on journal recommended structure, this section should be moved to the Methods section. Also, the authors self-cited their own book chapter too many times in this paper. Since book chapters do not usually go through the peer-review process, authors should make an effort in this paper to explain this model in details and avoid too many self-citations.

5. Evidence supporting SCIAM: This section should be moved to the discussion and the comparison between the findings of this paper and previous work presented.

6. Methods:

Subsection 6.2: The authors extensively describe the many factors that impact adjustment (Section 3); yet either none of these factors were assessed in the models described in 6.2 or the link between these and SCIM and Secondary Measures was not adequately described. This causes the manuscript to read as two distinct articles rather than one comprehensive manuscript.

7. Results

In 7.1 subsection, Model 1, first and second paragraphs are almost identical with minor differences (positive vs negative) and reporting the numbers shown in the tables and figures. Authors should make an effort in describing the relationships between different factors based on the statistical findings rather than just reporting numbers (which readers can already see in the tables/figures) and making the description of the findings more engaging for the readers. In addition, the distinction between the two paragraphs (other than the values) is not made until the last sentence. It is suggested that the first sentence of each paragraph states up front that the first paragraph refers to positive vitality/mental health and the second paragraph to negative vitality/mental health or that both paragraphs be merged into one to describe the differences/similarities of the positive and negative factors. Please combine table 2 and 3 and Figures 3 and 4 (panels a and b) to make the visual/numeric comparison easier for the reader.

Subsection 7.2 Model 2: same comment as 7.1. Also, combine figure 5 and 6 and table 4 and 5.

Discussion:

Discussion should include comparisons between the findings of this study with previous studies and the differences/similarities with previous work discussed. Also possible connections/comparison between model 1 and model 2 should be presented. The authors state that: “For example, although the data was not presented, the influence of the moderator “social participation” on adjustment was also found to be significantly influenced by mediation factors. Equally, other mediator/appraisal factors could have been selected from the available database and tested in the model analyses.” Why aren’t all these comparisons/data reported in this paper? Authors are encouraged to perform a comprehensive analysis of their data and present all the results in the results section or as supplementary.

Reviewer 2 Report

I reviewed the article entitle"Investigating the Complex Dynamics of Adjustment for Individuals with Spinal Cord Injury: A Narrative Review and Mediation Model Analysis" as a review article . However, I found it confusable. It is not neither a review article nor original one. It doesn't follow the right path as a review article. authors didn't describe how mediation model analysis work.  Is there any software specified for this kind of analysis. How they interpret the obtain significancy.  I really didn't understand what exactly authors want to share with their audiance.In line 222 authors claim that pain and depression have a positive association. But in general it is not correct. Chronic pain may increase depressive mood but vice versa is not applicable.

Round 2

Reviewer 2 Report

with this revision, it is now acceptable.